# The Influence of Body Mass Index on Growth Factor Composition in the Platelet-Rich Plasma in Patients with Knee Osteoarthritis

**DOI:** 10.3390/ijerph20010040

**Published:** 2022-12-20

**Authors:** Michał Wiciński, Dawid Szwedowski, Łukasz Wróbel, Sławomir Jeka, Jan Zabrzyński

**Affiliations:** 1Department of Pharmacology and Therapeutics, Faculty of Medicine, Collegium Medicum, Nicolaus Copernicus University, M. Curie 9, 85-090 Bydgoszcz, Poland; 2Department of Organic Chemistry, Faculty of Pharmacy, Collegium Medicum, Nicolaus Copernicus University, Dr. A. Jurasza St. 2, 85-094 Bydgoszcz, Poland; 3Orthopedic Arthroscopic Surgery International (O.A.S.I.) Bioresearch Foundation, Gobbi N.P.O., 20133 Milan, Italy; 4Department of Rheumatology and Connective Tissue Diseases, University Hospital No. 2, Collegium Medicum, Nicolaus Copernicus University, 85168 Bydgoszcz, Poland; 5Department of General Orthopedics, Musculoskeletal Oncology and Trauma Surgery, University of Medical Sciences, 61-701 Poznan, Poland; 6 Department of Pathology, Faculty of Medicine, Collegium Medicum, Nicolaus Copernicus University, M. Curie 9, 85-090 Bydgoszcz, Poland

**Keywords:** knee osteoarthritis, growth factors, obesity, platelet-rich plasma, body mass index

## Abstract

Background: An abnormally high body mass index is strongly associated with knee osteoarthritis. Usually, obese patients are excluded from clinical trials involving PRP intra-articular injections. Growth factors have been demonstrated to have a disease-modifying effect on KOA treatment, even though data on their influence on treatment effectiveness in obese patients are lacking. Purpose: To prospectively compare the level of selected growth factors including transforming growth factor-b (TGF-β), epidermal growth factor (EGF), fibroblast growth factor, insulin-like growth factor-1 (IGF-1), platelet-derived growth factor (PDGF), vascular endothelial growth factor (VEGF), and fibroblast growth factor-2 (FGF-2) in platelet-rich plasma (PRP) in obese patients and patients with normal BMI. Methods: A total of 49 patients were included in the study according to inclusion and exclusion criteria. The groups strongly differed in body mass index (median values 21.6 vs. 32.15). Concentrations of growth factors were measured with an enzyme-linked immunosorbent assay. Statistical significance was determined with the Mann-Whitney U test. The compliance of the distribution of the results with the normal distribution was checked using the Shapiro–Wilk test separately for both groups. Results: There were no statistically significant differences in median marker levels between groups. Statistically significant Pearson correlations were observed between IGF-1 serum level and age (weak negative, r = −0.294, *p* = 0.041) and gender (moderate positive, r = 0.392, 0.005). Conclusions: BMI does not influence the level of selected growth factors in patients with knee osteoarthritis. Obese and non-obese patients had similar compositions of PDGF, TGF-β, EGF, FGF-2, IGF-1, and VEGF. PRP can be used in both groups with similar effects associated with growth factors’ influence on articular cartilage.

## 1. Introduction

Knee Osteoarthritis (KOA) is one of the most common diseases of the joints that leads to significant social and economic problems [1]. The pathogenesis of cartilage degeneration is multifactorial and the molecular pathways responsible for this process are still unknown [2,3]. Due to the complexity of the processes in the KOA, disease-modifying drugs are still missing [4]. Recently, growth factor therapy has become more and more popular in the treatment of musculoskeletal system disorders. Growth factors promote proliferation and angiogenesis, reducing critical inflammatory regulators and the expression of inflammatory enzymes [5,6]. Some “in vitro” and “in vivo” studies demonstrated that growth factors are responsible for improved cellular remodeling and decreased time to healing in knee joint cartilage [7,8]. Platelet-rich plasma (PRP) has been shown to contain and release growth factors wherein there are especially active factors: transforming growth factor-b (TGF-β), platelet-derived growth factor (PDGF), vascular endothelial growth factor (VEGF), and basic fibroblast growth factor (bFGF). PDGF is a potent chemotactic factor for cells of mesenchymal origin, on which it has a proliferative effect [9]. Insulin-like Growth Factor-1 (IGF-1) also plays an important role in muscle regeneration. It promotes both the proliferation and differentiation of myoblasts, protects against atrophy, and induces hypertrophy of muscle fibers [9]. Fibroblast growth factor regulates the normal development and homeostasis of articular cartilage, as evidenced by the fact that abnormal FGF signaling contributes to the onset and progression of osteoarthritis [10]. Vascular endothelial growth factor stimulates angiogenesis and vasculogenesis. Additionally, it initiates a macrophage-related angiogenic response in the inflammation stage [11]. Transforming growth factor-b plays an important role in cell growth and cellular differentiation to chondrocytes. It also inhibits collagen synthesis and calcium release which can prevent subchondral sclerosis.

A Body Mass Index (BMI) over 25 is one of the major risk factors for the development of knee osteoarthritis [12]. Usually, in clinical trials with PRP intra-articular injections, obese patients are excluded from the study [13,14]. Although there is still a lack of data regarding the influence of the growth factors on KOA treatment effectiveness, it was demonstrated that the growth factors can have a disease-modifying effect [15].

This study aimed to compare growth factor levels in platelet-rich plasma in KOA patients. We hypothesized that levels of growth factors could be influenced by BMI. Accordingly, we aimed to compare growth factor levels, searching for differences between patients with and without obesity.

## 2. Methods

### 2.1. Patient Selection and Screening

The study protocol was approved by an appropriate Institutional Review Board (KB 332/2021) before the enrolment of the first patient. The study was performed following the ethical standards outlined in the 2013 revision of the 1975 Declaration of Helsinki. Each participant signed written consent.

There were five inclusion criteria for patients: radiographic evidence of Kellgren–Lawrence grade 2 to grade 3, patient aged 40–75 years, a restriction on taking pain relievers, and a ban on taking paracetamol 24 h before the appointment.

Anemia, diabetes, coagulation disorders, or taking anticoagulants were among the exclusion criteria. Rheumatic diseases, systemic diseases of connective tissue, active neoplasms, oral steroids, antibiotics, or biological treatments were also exclusion criteria in the study. A combination of nonsteroidal anti-inflammatory drugs and chondroprotective supplements was not permitted during the trial. Study participants were allowed to take paracetamol during the study, but they had to discontinue it 48 h before each follow-up visit.

PRP was prepared by collecting small amounts of peripheral blood (10–12 mL) and centrifuging it (Zenithlab 80-2C, Zenith Lab Inc., Pomona, CA, USA). The PRP centrifuge was set to 4000 rpm and had a duration of 10 min. During centrifugation at 1500× *g*, the plasma fraction was separated from the rest of the peripheral blood, concentrating the platelets and producing 7–8 mL of leukocyte-poor PRP.

### 2.2. ELISA

Growth factors’ levels were determined with the ELISA method on a EPOCH microplate spectrophotometer (BioTech, Santa Clara, CA, USA) using SunRed ELISA kits (Sunredbio(SRB) Technology, Shanghai, China) for TGF-β, FGF-2, EGF, VEGF, PDGF, and DiaMetra kits (DiaMetra Srl Unipersonale, Spello, Italy) for IGF-1. Intra and inter-assay coefficients of variability (CV) are presented in the Table 1.

### 2.3. Statistical Methods

A total of 49 patients who completed the study were statistically analyzed and further analyzed by subgroups. Data analysis was performed with Statistica 13.3. The results for both groups (control—1 and study—2) were presented as median and mean values with standard error of the mean (±SEM). Statistical significance was determined with the Mann–Whitney U test. The compliance of the distribution of the results with the normal distribution was checked using the Shapiro–Wilk test separately for both groups (control—1 and study—2). None of them showed normal distribution. Values of *p* ≤ 0.05 were considered statistically significant.

Pearson correlation coefficient between basic anthropometric data such as BMI, age, gender, and markers for both groups and all patients was also calculated. Values of *p* ≤ 0.05 were considered statistically significant. Values of r ≤ 0.3 were considered as weak correlation, 0.3 < r ≤ 0.5 as moderate, and r > 0.5 as strong.

## 3. Results

We assessed 196 patients for eligibility from August 2021 to March 2022. Of these patients, 141 did not meet the inclusion criteria and 5 declined to take part. The study included 50 patients, 25 in each group. Forty-nine patients completed the study, as one patient did not report diabetes type II which was an exclusion criterion. In the obese group, the mean age was 52.5 years, while the average age in the control group was 62.17 years, Table 2.

The groups strongly differed in body mass index (median values 21.6 vs. 32.15). Figure 1 presents the box plot of BMI of both groups (1 - control. 2 - study). There were no statistically significant differences in median serum markers level between groups. The box plots of markers for both groups were presented in Figure 2.

Statistically significant Pearson correlations were observed between basic anthropometric data and part of blood serum markers: BMI vs. TGF-β in the control group (moderate negative correlation); age vs. IGF-1 in two groups combined (weak negative) and study group (moderate negative); age vs. FGF-2 in the study group (strong negative correlation); and gender vs. IGF-1 in groups combined (moderate positive) and study (strongly positive).

Strong positive statistically significant Person correlations were observed between all blood serum markers except IGF-1.

Statistically significant Pearson correlations were observed between IGF-1 serum level and age (weak negative, r = −0.294, *p* = 0.041) and gender (moderate positive, r = 0.392, 0.005).

## 4. Discussion

Obesity is the most important risk factor of KOA and is believed to be modifiable. It is characterized by chronic inflammation, as evidenced by the increase in circulating pro-inflammatory cytokines [16]. Most of the studies suggesting a complex role of these mediators have shown an age-gender relationship on serum growth factor levels [17,18,19,20]. There have already been some reports about higher angiogenic factors in overweight and obese patients, but no studies on the relationship between growth factor levels and body mass index have been published to our knowledge.

The impact of BMI on the course of KOA and its progression is unknown. In a meta-analysis by Reyes et al., increase in BMI was associated with a 35% increased risk of KOA diagnosed clinically and radiologically [21]. Another cohort study showed that a weight loss of >10% could reduce the clinical manifestation of symptomatic KOA [22]. Most of the studies showed that obesity remains the most determinant risk factor of KOA even though it is considered modifiable and the level of obesity is directly correlated with clinical symptoms [12]. It is one of the reasons why usually, patients with abnormal BMI are excluded from controlled clinical trials using intra-articular PRP injections [23]. Physical activity and a special diet can reduce the clinical consequences of KOA, however, it could not be enough to reduce pain and improve joint function. Another reason is that a higher BMI leads to dysregulation of pro-inflammatory cytokine patterns. Certain inflammatory enzymes like interleukin-2 (IL-2), interleukin-10 (IL-10), interferon-γ (IFN-γ), and interleukin 1-α (IL1-α) were found to differ between participants with obesity and with normal BMI [24]. In the study by Tucker et al. interleukin 6 (IL-6) levels correlated positively with BMI [25]. In addition, interleukin-8 (IL-8), was shown to be elevated in adults with obesity [26]. However, these pro-inflammatory enzymes are just currently considered as the KOA markers, but the correlation between onset and consequences is still unknown.

Differences in the PRP composition have been shown between different systems for obtaining platelet-rich plasma [19,27]. It was also suggested that variability in PRP composition may be due to the differences in the plasma composition of patients [28,29]. There is still a lack of studies on the differences in the composition of PRP between different people and different groups of people. This study specifically assessed interpersonal differences in PRP composition using a standard hematology protocol to generate leukocyte-depleted PRP. The variability in PRP composition is large and this variability complicates the determination of PRP’s mechanisms of action and clinical efficacy. It was demonstrated that high-diet and saturated fatty acid palmitate leads to decreased IGF-1–mediated proteoglycan production compared with a low-fat diet in a mice model [30]. Medications may also play a role. An in vitro study by dexamethasone is synergistic with increased IGF-1 levels in preventing cytokine-mediated chondrocyte damage [31]. Additionally, some diseases like diabetes mellitus could influence the growth factor levels in serum and they were included in the exclusion criteria. An interesting finding was presented by Caroleo et al. (2019), that EGF is associated with eating dysfunctional behaviors like anorexia nervosa [24].

This study aimed to describe the association between selected growth factors and BMI. Although the impact of BMI on KOA is unknown, a meta-analysis found that people with KOA were the least active [32]. Consequently, it is difficult for them to be physically active and the risk of disability is higher. Additionally, being obese is associated with a lower quality of life and may affect the clinical assessment of knee joints [33]. Intra-articular injections could be a reasonable option for moderate KOA in this group of patients to improve function and decrease pain. In vitro and in vivo studies have shown a positive effect of intra-articular PRP injections in knee OA, including improvement of cellular remodeling and a reduction of healing time. Although there are many ways to obtain PRP, it was shown to contain and release growth factors, including TGF-β, PDGF, VEGF, IGF-1, FGF-2, and EGF. Growth factors are a group of polypeptides that modify cell proliferation and positively affect articular cartilage. Despite concerns about growth factor analysis, more and more data are emerging about the personalized medicine approach in this treatment [34]. Some studies have looked at the relationship between age and the growth factors level in PRP. It was demonstrated that the levels of PDGF-BB, TGF-β1, IGF-1, and EGF contained in PRP were statistically higher in younger people (under the age of 25) [35]. Despite these findings, we did not find any significant differences in serum levels of TGF-β, PDGF, EGF, VEGF, and FGF-2 in our sample. Another study showed a small but significant negative correlation between IGF-1 levels and age, but no significant influence of age on PDGF or TGF-β. Our data confirmed these findings and demonstrated negative correlations between IGF-1 level and age (weak negative, r = −0.294, *p* = 0.041). However, a study by Cho et al. presented a weak negative correlation between age and growth factors [36]. The ratio of growth factors to inflammatory cytokines seems to be more useful to assess the clinical effectiveness and was positive in the younger group and negative in the older group, potentially suggesting a higher regenerative capacity in the young. These proportions are important to understand factors affecting the action of PRP in the knee joint. Further investigation of these inter-relations is required. Xiong et al., presented significant differences in the composition of PRP between genders in healthy patients [37]. PRP obtained from men contained a higher level of PDGF-BB, VEGF, and TGF-β1 compared with women. Age-related differences were less pronounced, with PRP from older patients showing lower IGF-1 content. In addition, significant within-group variability in the PRP composition was observed. In our study, we found a relationship between gender and growth level only for IGF-1. Age was correlated with IGF-1 and FGF-2 in the study group (Table 3). We also observed high inter-relations between selected growth factors (Table 4). Some studies demonstrated differences between PRP composition in different methods of obtaining [38,39] (Table 5).

We excluded patients with rheumatoid diseases and other autoimmunological disorders because the levels of growth factors could be influenced by an inflammatory response and immune system. No significant differences were found between groups with and without obesity, but previous findings on the influence of age and gender on growth factors were confirmed.

Since complex interactions occur between cytokines, metalloproteinases, and growth factors in the pathogenesis of KOA, more complex analyses comparing patients with different BMI are needed to find the factors responsible for a different course of illness in these groups.

PRP was only obtained from patients with KOA in this study, which is a limitation. We have not obtained the PRP from healthy volunteers as a control group. However, PRP is typically used for the relief of knee osteoarthritis, in patients who may have comorbidities such as obesity.

Although the whole blood collection was standardized, a previously described freeze-thaw method of obtaining PRP for analysis may not reflect in vivo conditions. Finally, we used a standard PRP preparation protocol as defined in hematology as opposed to a commercial system. Our study reduces the risk of bias due to the extensive inclusion and exclusion criteria for this study. Future analyzes of the role of growth factors in KOA patients should be extended to other molecules such as cytokines and metalloproteinases.

## 5. Conclusions

Growth factors TGF-β, EGF, IGF-1, PDGF, VEGF, and FGF-2 were included in the study because of their importance in the KOA treatment. This exploratory study compared these growth factors’ levels, both in obese and normal KOA patients. We also considered age, sex, and internal relations between growth factors as potential confounding variables. Only levels of IGF-1 were influenced by age and gender. This study evaluated differences in the PRP composition using a standard hematology protocol to generate PRP. Further investigation should take into account that variability in the composition of PRP is high; the interplay between growth factors, proteinases, and cytokines is complex and still unknown; and these issues complicate the determination of the mechanisms of action of PRP and the role of selected growth factors.

## Figures and Tables

**Figure 1 ijerph-20-00040-f001:**
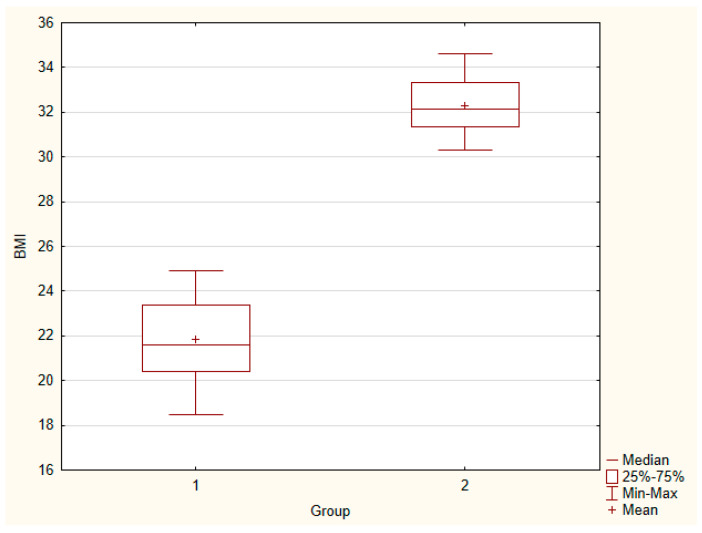
Body mass index of control group (1) and study group (2). U = 0.00, *p* = 0.000.

**Figure 2 ijerph-20-00040-f002:**
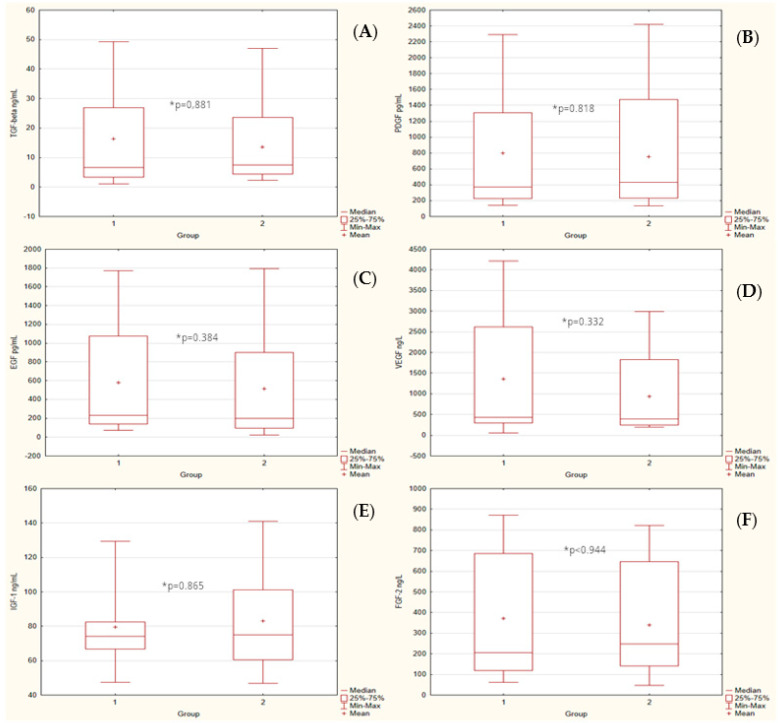
Concentration of markers of control (1) and study group (2). (**A**) TGF-β, U = 292, * *p* = 0.881; (**B**) PDGF, U = 288, * *p* = 0.818; (**C**) EGF, U = 256, * *p* = 0.384; (**D**) VEGF, U = 251, * *p* = 0.332; (**E**) IGF-1, U = 291, * *p* = 0.865; and (**F**) FGF-2, U = 296, * *p* < 0.944.

**Table 1 ijerph-20-00040-t001:** Intra- and inter-assay coefficients of variability.

Biomarker	Kit Name	Intra Assay CV	Inter Assay CV
Insulin-like growth factor 1	Quantitative immunoenzymatic determination of human inuslin-like growth factor 1 (IGF-1) in human serum	≤8.9%	≤12.9%
Transforming growth factor beta	Human transforming growth factor β (TGF-β) ELISA kit	<10%	<12%
Fibroblast growth factor 2	Human fibroblast growth factor 2 (FGF2) ELISA kit	<10%	<12%
Epidermal growth factor	Human epidermal growth factor 2 (EGF) ELISA kit	limited data	limited data
Vascular endothelial growth factor	Human vascular endothelial cell growth factor (VEGF) ELISA kit	<10%	<12%
Platelet-derived growth factor	Human platelet-derived growth factor (PDGF) ELISA Kit	<10%	<12%

**Table 2 ijerph-20-00040-t002:** Anthropometric data of patients.

		Control Group	Study Group
Male	Number	6	4
	Age [mean]	62.17	52.50
	BMI [mean][kg/m^2]	23.60	33.35
Female	Number	19	20
	Age [mean]	56.58	54.10
	BMI [mean][kg/m^2]	21.31	32.08
Both genders	Number	25	24
	Age [mean]	57.92	53.83
	BMI [mean][kg/m^2]	21.86	32.29

**Table 3 ijerph-20-00040-t003:** Statistical data for both groups (1 - control and 2 - study): mean ± SEM, median, and U Mann-Whitney test’s U and *p* values.

	Mean ± SEM (1)	Mean ± SEM (2)	Median (1)	Median (2)	U	*p*-Value
TGF-β [ng/mL]	16.41 ± 3.27	13.60 ± 2.75	6.58	7.55	292.00	0.881
PDGF [pg/mL]	801.05 ± 148.68	756.27 ± 146.31	372.06	433.28	288.00	0.818
EGF [pg/mL]	578.31 ± 110.25	514.43 ± 121.1	231.66	196.50	256.00	0.384
VEGF [ng/L]	1360.81 ± 288.44	939.03 ± 206.46	425.46	395.40	251.00	0.332
IGF-1 [ng/mL]	79.38 ± 4.27	82.97 ± 5.67	74.06	75.09	291.00	0.865
FGF-2 [ng/L]	372.58 ± 61.49	338.84 ± 53.13	204.50	246.93	296.00	0.944
BMI	21.86 ± 0.42	32.29 ± 0.28	21.60	32.15	0.00	0.000

**Table 4 ijerph-20-00040-t004:** Pearson correlation between markers and basic anthropometric data.

		All	Control	Study
BMI	TGF-β	r = −0.1715 *p* = 0.239	r = −0.4165 *p* = 0.038	r = 0.0484 *p* = 0.822
	PDGF	r = −0.0889 *p* = 0.543	r = −0.3068 *p* = 0.136	r = 0.0025 *p* = 0.991
	EGF	r = −0.1173 *p* = 0.422	r = −0.3747 *p* = 0.065	r = 0.0446 *p* = 0.836
	VEGF	r = −0.2326 *p* = 0.108	r = −0.3484 *p* = 0.088	r = 0.0424 *p* = 0.844
	IGF-1	r = 0.1136 *p* = 0.437	r = 0.2121 *p* = 0.309	r = 0.0659 *p* = 0.760
	FGF-2	r = −0.1018 *p* = 0.487	r = −0.1932 *p* = 0.355	r = −0.0463 *p* = 0.830
Age	TGF-β	r = −0.1088 *p* = 0.457	r = −0.0794 *p* = 0.706	r = −0.2095 *p*=0.326
	PDGF	r = −0.1917 *p* = 0.187	r = −0.1592 *p* = 0.447	r = −0.2561 *p* = 0.227
	EGF	r = −0.1673 *p* = 0.251	r = −0.0796 *p* = 0.705	r = −0.2949 *p* = 0.162
	VEGF	r = −0.0892 *p* = 0.542	r = −0.0890 *p* = 0.672	r = −0.2065 *p* = 0.333
	IGF-1	r = −0.2937 *p* = 0.041	r = −0.1149 *p* = 0.585	r = −0.4364 *p* = 0.033
	FGF-2	r = −0.2589 *p* = 0.072	r = −0.1028 *p* = 0.625	r = −0.5133 *p* = 0.010
Gender	TGF-β	r = −0.2096 *p* = 0.148	r = −0.1814 *p* = 0.386	r = −0.2766 *p* = 0.191
	PDGF	r = −0.2145 *p* = 0.139	r = −0.1309 *p* = 0.533	r = −0.3273 *p* = 0.118
	EGF	r = −0.2171 *p* = 0.134	r = −0.1568 *p* = 0.454	r = −0.3003 *p* = 0.154
	VEGF	r = −0.1842 *p* = 0.205	r = −0.1529 *p* = 0.466	r = −0.2924 *p* = 0.166
	IGF-1	r = 0.3924 *p* = 0.005	r = 0.2688 *p* = 0.194	r = 0.5395 *p* = 0.007
	FGF-2	r = −0.2466 *p* = 0.088	r = −0.1864 *p* = 0.372	r = −0.3485 *p* = 0.095

**Table 5 ijerph-20-00040-t005:** Pearson correlation between markers.

		All	Control	Study
TGF-β	PDGF	r = 0.8848 *p* = 0.000	r = 0.8276 *p* = 0.000	r = 0.9698 *p* = 0.000
	EGF	r = 0.9533 *p* = 0.00	r = 0.9591 *p* = 0.000	r = 0.9661 *p* = 0.000
	VEGF	r = 0.9749 *p* = 0.00	r = 0.9820 *p* = 0.000	r = 0.9769 *p* = 0.000
	IGF-1	r = −0.0002 *p* = 0.999	r = 0.1642 *p* = 0.433	r = −0.1461 *p* = 0.496
	FGF-2	r = 0.7754 *p* = 0.000	r = 0.9059 *p* = 0.000	r = 0.5779 *p* = 0.003
PDGF	EGF	r = 0.9098 *p* = 0.00	r = 0.8493 *p* = 0.000	r = 0.9736 *p* = 0.000
	VEGF	r = 0.8646 *p* = 0.000	r = 0.8235 *p* = 0.000	r = 0.9748 *p* = 0.000
	IGF-1	r = 0.0161 *p* = 0.912	r = 0.2007 *p* = 0.336	r = −0.1327 *p* = 0.537
	FGF-2	r = 0.7051 *p* = 0.000	r = 0.7872 *p* = 0.000	r = 0.6029 *p* = 0.002
EGF	VEGF	r = 0.9461 *p* = 0.00	r = 0.9736 *p* = 0.000	r = 0.9725 *p* = 0.000
	IGF-1	r = 0.0282 *p* = 0.847	r = 0.1516 *p* = 0.469	r = −0.0558 *p* = 0.796
	FGF-2	r = 0.7662 *p* = 0.000	r = 0.8866 *p* = 0.000	r = 0.8866 *p* = 0.000
VEGF	IGF-1	r = 0.0218 *p* = 0.882	r = 0.1916 *p* = 0.359	r = −0.1406 *p* = 0.512
	FGF-2	r = 0.7855 *p* = 0.000	r = 0.9135 *p* = 0.000	r = 0.5767 *p* = 0.003
IGF-1	FGF-2	r = 0.1375 *p* = 0.346	r = 0.0988 *p* = 0.639	r = 0.1905 *p* = 0.373

## Data Availability

Not Applicable.

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
