# Peer review of "The Influence of Body Mass Index on Growth Factor Composition in the Platelet-Rich Plasma in Patients with Knee Osteoarthritis"

_ijerph, 2022, doi:10.3390/ijerph20010040_

Round 1
Reviewer 1 Report
The authors presented an interesting prospective single-center study showing the levels of selected growth factors in obese patients with knee osteoarthritis that were treated with platelet-rich plasma. The presented study shows the clinical implications of PRP treatment in obese patients with knee osteoarthritis, but the clinical usefulness should be confirmed by a multi-center study involving more patients.
The Introduction sufficiently presents the background for the study.
The biggest Material and Methods deficiency is the lack of PRP preparation method (and maybe the therapy duration/injections frequency etc). This aspect of the study is completely missing. In the Discussion, you write: “ Differences in the PRP composition have been shown between different systems for 181 obtaining platelet-rich plasma [19][27].”, so it is evident that the method of PRP preparation matters. Please include the PRP preparation method in Materials and Methods.
Also, please provide the names and country of origin for all used reagents and instruments as well as the intra- and inter assay coefficients of variability for ELISA kits (together with their full names) in Materials and Methods section.
The presentation of the results could use some editing – please consult the journal guidelines. Also, please make sure that the Tables and Figures captions are self-explanatory as in the current version they do not comply with that principle.
The Discussion is also well written. It contains a paragraph about the paper's limitations.
The conclusions paragraph should be shortened and refer only to the obtained results. Any other information about the study background could be used in Abstract which is missing the background part.
Final decision: minor revision
Author Response
Dear Reviewer,
We are grateful for the Your thoughtful comments. We agree with most of the recommendations and revised our paper according to them. All changes within the manuscript are corrected with track changes. The corrections and revisions are as follows:
The authors presented an interesting prospective single-center study showing the levels of selected growth factors in obese patients with knee osteoarthritis that were treated with platelet-rich plasma. The presented study shows the clinical implications of PRP treatment in obese patients with knee osteoarthritis, but the clinical usefulness should be confirmed by a multi-center study involving more patients.
The Introduction sufficiently presents the background for the study.
The biggest Material and Methods deficiency is the lack of PRP preparation method (and maybe the therapy duration/injections frequency etc). This aspect of the study is completely missing. In the Discussion, you write: “ Differences in the PRP composition have been shown between different systems for 181 obtaining platelet-rich plasma [19][27].”, so it is evident that the method of PRP preparation matters. Please include the PRP preparation method in Materials and Methods.
Response: Thank you for your comment. We added the information about the preparation method of PRP in Materials and Methods section. We haven’t used this PRP for the intra-articular injection. It was used only for diagnostic purposes.
Also, please provide the names and country of origin for all used reagents and instruments as well as the intra- and inter assay coefficients of variability for ELISA kits (together with their full names) in Materials and Methods section.
Response: Thank you for this comment. We added the information about the intra- and inter assay coefficients of variability for ELISA kits (together with their full names in the Table 1.
The presentation of the results could use some editing – please consult the journal guidelines. Also, please make sure that the Tables and Figures captions are self-explanatory as in the current version they do not comply with that principle.
Response: We edited the results section according to the journal guidelines.
The Discussion is also well written. It contains a paragraph about the paper's limitations.
The conclusions paragraph should be shortened and refer only to the obtained results. Any other information about the study background could be used in Abstract which is missing the background part.
Response: Thank you for your recommendation. We agree with Your comment. We removed some parts of the conclusion and added the background paragraph to the abstract.
Kind regards,
Dawid Szwedowski
Reviewer 2 Report
I find the article appropriate and the subject matter interesting. However, the article should describe more adequately the methods and the results obtained. The presentation of the tables and graphs should be more careful.
· Introduction. Line 46. Please, put "in vitro" and "in vivo" in cursive.
· Introduction. Line 53. Please, define IGF-1, as first mentioned in the main text.
· Introduction. Lines 62-63. The text is cut off.
· Introduction. Lines 68. The citation [12] is of for the previous line.
· Introduction. Lines 68-69. "Besides the elevated pressure on the articular cartilage, it can influence the immune system". Please, cite properly.
· Introduction. Lines 68-69. The citation [13] is not about a clinical trial, but the text reference this study. Please amend it. "Usually in clinical trials with PRP intra-articular injections obese patients were excluded from the study [13][14]."
· Methods. Line 96. Point 2.2. ELISA: Please, specify the molecules analysed.
· Methods. The obtaining methods for PRP obtaining are missing. It is mandatory the detailed description of the methods.
· Results. Table 1. Anthropometric data of patients. The authors should clarify if the BMI indicated in this table is mean or median.
· Results. Table 2. Statistical data for both groups. Please clarify the statistical test used, and groups 1 and 2 more visual.
· Results. Table 3. Pearson correlation between markers and basic anthropometric data. The authors must indicate the "r", in the same way that the p value. For example, r= -0.1715
· Results. Table 4. The same as for table 3.
· Results. Diagram 1 and 2. Indicate graphically, with *, the significance inside each graph.
· Discussion. Line 182. "that th variability" Please, correct this typo.
· Discussion. Line 187 "PRP.45 The "Again, correct this typo.
· References: Please check ALL references for accurate content and format. For example, ref # 34 has neither volume nor pages. Complete please: https://pubmed.ncbi.nlm.nih.gov/30246605.
Author Response
Dear Reviewer,
We are grateful for Your thoughtful comments. We agree with most of the recommendations and revised our paper according to them. All changes within the manuscript are corrected with track changes. The corrections and revisions are as follows:
I find the article appropriate and the subject matter interesting. However, the article should describe more adequately the methods and the results obtained. The presentation of the tables and graphs should be more careful.
Response: Thank You for Your comment. We edited the methods and results section according to the journal guidelines.
- Introduction. Line 46. Please, put "in vitro" and "in vivo" in cursive.
- Introduction. Line 53. Please, define IGF-1, as first mentioned in the main text.
- Introduction. Lines 62-63. The text is cut off.
- Introduction. Lines 68. The citation [12] is of for the previous line.
- Introduction. Lines 68-69. "Besides the elevated pressure on the articular cartilage, it can influence the immune system". Please, cite properly.
- Introduction. Lines 68-69. The citation [13] is not about a clinical trial, but the text reference this study. Please amend it. "Usually in clinical trials with PRP intra-articular injections obese patients were excluded from the study [13][14]."
- Methods. Line 96. Point 2.2. ELISA: Please, specify the molecules analysed.
- Methods. The obtaining methods for PRP obtaining are missing. It is mandatory the detailed description of the methods.
- Results. Table 1. Anthropometric data of patients. The authors should clarify if the BMI indicated in this table is mean or median.
- Results. Table 2. Statistical data for both groups. Please clarify the statistical test used, and groups 1 and 2 more visual.
- Results. Table 3. Pearson correlation between markers and basic anthropometric data. The authors must indicate the "r", in the same way that the p value. For example, r= -0.1715
- Results. Table 4. The same as for table 3.
- Results. Diagram 1 and 2. Indicate graphically, with *, the significance inside each graph.
- Discussion. Line 182. "that th variability" Please, correct this typo.
- Discussion. Line 187 "PRP.45 The "Again, correct this typo.
- References: Please check ALL references for accurate content and format. For example, ref # 34 has neither volume nor pages. Complete please: https://pubmed.ncbi.nlm.nih.gov/30246605.
Response: Thank you for Your valuable recommendations. We agreed with Your comments and addressed all of them.
I appreciate You again for Your thoughtful criticism and kind advice. Your review greatly helped to refine our manuscript.
With best regards,
Dawid Szwedowski
Round 2
Reviewer 2 Report
· Line 40. Please correct typo: 2TGF-βTGF-β"
· Lines 102-103: The authors state "The PRP centrifuge was set to 4000 rpm and had a duration of 10 min." But in order to compare protocols the autors must provide the centrifugation speed in "g" values. Thank you. Please, classify the PRP (there are multiple classifications of PRP) indicating whether it is an L-PRP or P-PRP-
· Again (as previous review), in the introduction. The citation [13] is not about a clinical trial, but the text reference this study. Please amend it. "Usually in clinical trials with PRP intra-articular injections obese patients were excluded from the study [13][14]"
· Results. Diagram 1 and 2. Indicate graphically, with *, the significance inside each graph. But, please, * ONLY in statistically significant values.
· Again (as previous review), References: Please check ALL references for accurate content and format. For example, ref # 34 has neither volume nor pages. Complete please: https://pubmed.ncbi.nlm.nih.gov/30246605.
I find the article adequate and the topic of interest. However, the article should more adequately describe the methods and the results obtained.
In addition, the authors have NOT satisfactorily answered the reviewer's requests. They have not taken the time to make easy and reasonable changes, such as completing the references or other minor changes.
Therefore, the authors should revisit my previous review report (and the actual one) and respond point by point to everything that was reasonably requested. The quality of the paper is compromised by these errors, and by extrapolation, that of the journal.
Author Response
December 15, 2022
Dear Reviewer,
Manuscript. Number.: ijerph-1990706
Title: The influence of Body Mass Index on the Growth Factors Composition in Platelet-Rich Plasma in Patients with Knee Osteoarthritis.
We are grateful for the thoughtful comments. We agree with all the recommendations and revised our paper according to them. All changes within the manuscript are corrected with track changes. In this letter, our answers were marked as blue and the detailed changes in the manuscript were marked as underlined blue. The corrections and revisions are as follows:
Line 40. Please correct typo: 2TGF-βTGF-β"
Response: Thank you for this comment. We corrected this typo.
Lines 102-103: The authors state "The PRP centrifuge was set to 4000 rpm and had a duration of 10 min." But in order to compare protocols the autors must provide the centrifugation speed in "g" values. Thank you.
Response: Thank you for this suggestion. We added the information about the centrifugation speed in ‘’g’’ values.
Please, classify the PRP (there are multiple classifications of PRP) indicating whether it is an L-PRP or P-PRP-
Response: Thank you for this suggestion. We added the information that we used leukocyte-poor PRP for further analysis.
Again (as previous review), in the introduction. The citation [13] is not about a clinical trial, but the text reference this study. Please amend it. "Usually in clinical trials with PRP intra-articular injections obese patients were excluded from the study [13][14]"
Response: Thank you for this comment. We changed this reference according to your suggestions.
Results. Diagram 1 and 2. Indicate graphically, with *, the significance inside each graph. But, please, * ONLY in statistically significant values.
Response: We changed the diagrams and indicated the significance with * only in statistically significant values.
Again (as previous review), References: Please check ALL references for accurate content and format. For example, ref # 34 has neither volume nor pages. Complete please: https://pubmed.ncbi.nlm.nih.gov/30246605.
Response: Thank you for this comment. We changed the references according to your suggestions.
I find the article adequate and the topic of interest. However, the article should more adequately describe the methods and the results obtained.
In addition, the authors have NOT satisfactorily answered the reviewer's requests. They have not taken the time to make easy and reasonable changes, such as completing the references or other minor changes.
Therefore, the authors should revisit my previous review report (and the actual one) and respond point by point to everything that was reasonably requested. The quality of the paper is compromised by these errors, and by extrapolation, that of the journal.
Response: Thank you for Your valuable comments. We believe that we answered and addressed all of the requests and we hope that after these changes the article is eligible for the Special Issue.
I appreciate You again for Your thoughtful criticism and kind advice. Your review greatly helped to refine our manuscript.
With best regards,
Dawid Szwedowski M.D., Ph.D.